# Vitamin D and Muscle Status in Inflammatory and Autoimmune Rheumatic Diseases: An Update

**DOI:** 10.3390/nu16142329

**Published:** 2024-07-19

**Authors:** Elvis Hysa, Emanuele Gotelli, Rosanna Campitiello, Sabrina Paolino, Carmen Pizzorni, Andrea Casabella, Alberto Sulli, Vanessa Smith, Maurizio Cutolo

**Affiliations:** 1Laboratory of Experimental Rheumatology and Academic Division of Clinical Rheumatology, Department of Internal Medicine, University of Genoa, Viale Benedetto XV, 6, 16132 Genoa, Italy; elvis.hysa@gmail.com (E.H.); campitiellorosanna@gmail.com (R.C.); sabrina.paolino@unige.it (S.P.); carmen.pizzorni@unige.it (C.P.); albertosulli@unige.it (A.S.); 2Department of Experimental Medicine (DIMES), University of Genoa, Via Leon Battista Alberti 2, 16132 Genoa, Italy; 3IRCCS Ospedale Policlinico San Martino, Largo Rosanna Benzi 10, 16132 Genoa, Italy; andrea.casabella@hsanmartino.it; 4Department of Internal Medicine, Ghent University, 9000 Ghent, Belgium; vanessa.smith@ugent.be; 5Department of Rheumatology, Ghent University Hospital, Corneel Heymanslaan 10, 9000 Ghent, Belgium; 6Unit for Molecular Immunology and Inflammation, VIB Inflammation Research Center (IRC), Technologiepark-Zwijnaarde 71, 9052 Ghent, Belgium

**Keywords:** vitamin D, connective tissue diseases, myositis, autoimmune rheumatic diseases

## Abstract

**Background and Objectives**: Vitamin D is a secosteroid hormone essential for calcium homeostasis and skeletal health, but established evidence highlights its significant roles also in muscle health and in the modulation of immune response. This review aims to explore the impact of impaired vitamin D status on outcomes of muscle function and involvement in inflammatory and autoimmune rheumatic diseases damaging the skeletal muscle efficiency both with direct immune-mediated mechanisms and indirect processes such as sarcopenia. **Methods**: A comprehensive literature search was conducted on PubMed and Medline using Medical Subject Headings (MeSH) terms: “vitamin D, muscle, rheumatic diseases.” Additionally, conference abstracts from The European Alliance of Associations for Rheumatology (EULAR) and the American College of Rheumatology (ACR) (2020–2023) were reviewed, and reference lists of included papers were scanned. The review emphasizes the evidence published in the last five years, while also incorporating significant studies from earlier years, structured by the extent of evidence linking vitamin D to muscle health in the most commonly inflammatory and autoimmune rheumatic diseases encountered in clinical practice. **Results**: Observational studies indicate a high prevalence of vitamin D serum deficiency (mean serum concentrations < 10 ng/mL) or insufficiency (<30 ng/mL) in patients with idiopathic inflammatory myopathies (IIMs) and polymyalgia rheumatica, as well as other autoimmune connective tissue diseases such as rheumatoid arthritis (RA), systemic lupus erythematosus (SLE) and systemic sclerosis (SSc). Of note, vitamin D insufficiency may be associated with reduced muscle strength (2 studies on RA, 2 in SLE and 1 in SSc), increased pain (1 study on SLE), fatigue (2 studies on SLE), and higher disease activity (3 studies on IIMs and 1 on SLE) although there is much heterogeneity in the quality of evidence and different associations for the different investigated diseases. Therefore, linked to the multilevel biological intervention exerted by vitamin D, several translational and clinical studies suggest that active metabolites of this secosteroid hormone, play a role both in reducing inflammation, but also in enhancing muscle regeneration, intra-cellular metabolism and mitochondrial function, although interventional studies are limited. **Conclusions**: Altered serum vitamin D status is commonly observed in inflammatory and autoimmune rheumatic diseases and seems to be associated with adverse muscle health outcomes. While maintaining adequate serum vitamin D concentrations may confer muscle-protective effects, further research is needed to confirm these findings and establish optimal supplementation strategies to obtain a safe and efficient serum threshold.

## 1. Introduction

Vitamin D is a steroid hormone which plays a crucial role in maintaining calcium homeostasis and skeletal health. Its synthesis is initiated in the skin upon exposure to ultraviolet rays (UV-B), followed by sequential hydroxylation in the liver and kidneys to produce the biologically active form, 1,25-dihydroxyvitamin D (1,25(OH)_2_D) [1].

Although vitamin D is traditionally acclaimed for its central role in regulating plasma concentrations of calcium and phosphate preserving a healthy mineralized skeleton, it displays many extra-skeletal functions. Recent evidence has shown its key role in the modulation of the immune system and inflammatory responses highlighting its protective character in many immune-related disorders [2,3].

The immunomodulatory role of this hormone extends from infections to autoimmune diseases. Different observational studies have shown an inverse relationship between vitamin D status and the risk of conditions such as acute respiratory infections, including coronavirus disease-19 (COVID-19), but also autoimmune diseases including rheumatoid arthritis (RA), multiple sclerosis, and type 1 diabetes [4,5,6,7].

In its biologically active form, vitamin D can modulate the activities of both innate and adaptive immune cells, including monocytes, macrophages, dendritic cells, B cells, and T cells. These cells express the vitamin D receptor (VDR), a nuclear receptor that binds to vitamin D response elements (VDRE) on DNA, regulating the transcription of genes involved in the immune response [8]. Specifically, vitamin D shifts gene expression from pro-inflammatory molecules like interleukin (IL)-1, IL-8, IL-12, tumor necrosis factor alpha (TNFα), interferon-γ, and toll-like receptors (TLR)-2 and 4 to anti-inflammatory cytokines such as IL-4, IL-5, and IL-10. Additionally, vitamin D promotes a regulatory phenotype in macrophages (M2) in the innate immune system and regulatory T and B cells in the adaptive immune system [2].

Vitamin D status has been also found to have a significant association with muscle health. Many studies have consistently shown that vitamin D deficiency (plasma concentrations < 10 ng/mL) or insufficiency (plasma concentrations < 30 ng/mL) are linked to various negative effects on muscle health including the loss of muscle strength, musculoskeletal pain, sarcopenia and increased risk of falls [9,10]. The body of evidence suggests that this hormone might exert a protective role on muscle health not only in terms of muscle regeneration and mitochondrial health but also by exhibiting anti-inflammatory properties [11,12]. Additionally, by maintaining calcium homeostasis, vitamin D is also involved in the regulation of muscle contraction. Indeed, recent evidence has shown that vitamin D insufficiency alters muscle contraction kinetics by reducing calcium reuptake into the sarcoplasmic reticulum, leading to a prolongation of the relaxation phase of muscle contraction [13].

The anti-inflammatory properties mediated by vitamin D in the skeletal muscle might be mediated by different mechanisms: through the upregulation of nuclear erythroid 2-related factor 2 (Nrf2), a transcription factor that regulates anti-oxidant and anti-inflammatory genes [14], by inducing a reduction of the oxidative stress [15] and by increasing the local production of the immunomodulatory interleukin (IL)-10. Additionally, this hormone has been shown to promote skeletal muscle regeneration after injury by regulating the function of satellite cells and enhancing mitochondrial health [11,16]. Further indirect evidence of muscle-related protective functions is provided by studies highlighting an upregulation of the vitamin D receptor (VDR) in the injured muscle indicating its role in the muscle repair process [16].

Considering that in many inflammatory rheumatic diseases the skeletal muscle is involved both by inflammation itself and/or by secondary sarcopenia related to overall disease activity, we aimed to explore whether vitamin D insufficiency or deficiency might have a role in these mechanisms.

Therefore, the review provides an update on the immune-modulatory properties of vitamin D in inflammatory and autoimmune rheumatic diseases involving the muscle health both with direct (i.e., immune-mediated damage) and/or indirect mechanisms (i.e., sarcopenia, reduced use).

## 2. Methods

A literature search was performed on PubMed and Medline using these Medical Subject Headings (MeSH) terms: “vitamin D, muscle, rheumatic diseases”.

The archive of the conference abstracts of The European Alliance of Associations for Rheumatology (EULAR) and American College of Rheumatology (ACR) of the last three years (2020–2023) was reviewed as well for pertinent publications. Additionally, the reference list of the included papers was scanned for additional publications meeting this study’s aim. When papers reported data partially presented in previous articles, we referred to the most recent published data.

The results of the search are reported as a narrative review with a focus on the most recent evidence published in the last 5 years, while also incorporating and discussing significant studies from earlier years.

The review is structured based on the extent of evidence linking vitamin D to muscle health in various inflammatory and autoimmune rheumatic diseases. We begin with diseases that directly affect muscles through immune-mediated mechanisms and conclude with conditions that impact muscles indirectly, such as through reduced muscle use due to overall disease activity and secondary sarcopenia.

The overall evidence is graphically represented in Figure 1.

## 3. Vitamin D and Idiopathic Inflammatory Myopathies

Idiopathic inflammatory myopathies (IIMs) represent a group of autoimmune muscle disorders characterized by chronic inflammation leading to progressive muscle weakness. These conditions, including dermatomyositis (DM), polymyositis (PM), and necrotizing myopathy, present a significant clinical burden, often persisting despite pharmacological interventions aimed at suppressing inflammation [29].

Vitamin D deficiency has emerged as a potential contributory factor in their pathophysiology (Table 1) and, in this respect, the pioneering observational study was written by Azali et al. in 2013 who detected that IIMs patients showed significantly lower serum concentrations of 25(OH) vitamin D compared to age- and sex-matched healthy controls (16 vs. 27 ng/mL, *p* = 0.0001) [17]. The authors concluded that vitamin D deficiency might be prevalent in patients with myositis and may act as a risk factor for developing IIMs in the adult, similarly to other autoimmune diseases [17].

A more recent study, also providing further immune-pathophysiological insights about vitamin D deficiency in IIMs patients, showed also significant correlations with the plasma muscle enzymes, the presence of anti-Jo-1 and anti-Mi-2 antibodies, and the absolute numbers of peripheral regulatory T cells in IIMs which were higher with the increasing serum concentrations of 25(OH) vitamin D [18]. From a clinical perspective, IIMs patients with reduced vitamin D serum concentrations of this cohort were more likely to have the heliotrope rash, gastrointestinal and liver involvement [18].

A former article had, instead, shown an inverse correlation between serum 25(OH) vitamin D and disease activity, assessed by physician global assessment (PGA), also in patients with juvenile DM (jDM) [30].

While no interventional studies of vitamin D in IIMs with outcomes on disease activity have been published, a single in-vitro study suggested that vitamin D might have a potential in modulating the inflammatory response of IIMs patients [19]. Indeed, the authors showed that the vitamin D receptor (VDR) agonist BXL-01-0029 reduced CXCL10 secretion in the human skeletal muscle via the downstream signaling of tumor necrosis factor alpha (TNFα). It was therefore suggested that this molecule might have a promising role in modulating inflammatory responses associated with IIMs considering that in their cohort CXCL10 concentrations were the highest in sera of subjects diagnosed with IIMs before therapy and was the only marker significantly different vs healthy controls [19].

Considering also the potential role of vitamin D in the overall skeletal muscle health, its insufficiency has been associated also with other adverse muscle health parameters in IIMs patients, including muscle damage, impaired regeneration, and altered energy metabolism.

In this respect, recent data presented at the EULAR Conference in 2022 showed that vitamin D signaling through the VDR is altered in the skeletal muscle of IIMs patients and is associated with disease activity and muscle function parameters, including myoglobin levels, creatine kinase (CK) concentrations and muscle strength tests scored by MMT8 [20]. Additionally, the decrease of VDR and CYP27B1 (the enzyme catalyzing conversion of calcifediol to calcitriol) gene expression with training suggest a potential role for vitamin D in the muscle response to exercise in IIMs [20].

Another report, published as an abstract form, showed the relationships between serum vitamin D concentrations with the expression of its receptor and parameters of lipid metabolism in cultured muscle cells of IIMs patients in-vitro suggesting the importance of vitamin D for muscle metabolic regulation (for more details see Table 1) [21].

Most of the aforementioned studies were very recently summarized by a meta-analysis which suggested that vitamin D deficiency might be both a risk factor for the development of IIMs, as a determinant of the immune dysregulation in this disease, or a consequence of the disease itself [31].

Interestingly, from a genetic perspective, a key investigation reported, among Hungarian patients with IIMs, that there were no differences in the VDR polymorphisms between them and healthy controls [32]. Conversely, a previous study published as an abstract from a conference held in UK had shown that a specific VDR polymorphism (rs2254210) was more frequently detected in DM patients with positive anti-155/140 antibodies (which target the transcription intermediary factor 1 [TIF1]) compared to the other subtypes of IIMs. Instead, no significant genetic associations were found in the other clinical sub-groups [21]. However, this study it is not specifically reported whether rs2254210 is a functional polymorphism or how it might affect VDR activity and vitamin D metabolism. Nevertheless, it is important to consider that certain VDR polymorphic variants may affect an individual’s ability to respond adequately to vitamin D, potentially leading to ‘vitamin D deficiency or insufficiency’ even when plasma levels of vitamin D are normal.

In summary, different observational and translational studies suggest that vitamin D insufficiency may play a concomitant role in the skeletal muscle damage of patients with IIMs both with mechanisms regulating the immune-inflammatory response and by the enhancement of favorable cellular metabolic pathways. Further longitudinal or randomized controlled studies might define if the supplementation of this hormone combined with the recommend immunosuppressive treatment might achieve a better control of the disease. 

**Table 1 nutrients-16-02329-t001:** Literature evidence on vitamin D status and idiopathic inflammatory myopathies (IIMs).

Author, Year	Patients	Controls	Results
**Significant associations**			
Robinson, 2012 [30]	N = 21	NA	Serum 25(OH)D levels were inversely associated with disease activity, assessed by PGA, in jDM
Azali, 2013 [17]	N = 149 IIMs including DM, PM, IBM and jDM	N = 290	Significantly lower serum levels of 25(OH) vitamin D than HCs (*p* = 0.0001). In IIMs the OR for deficient versus normal vitamin D concentrations was 17.7 (95% CI 8.1 to 38.6) and the OR for insufficient versus normal was 2.4 (95% CI 1.2 to 4.7).
Yu, 2021 [18]	N = 63 IIMs, including DM and PM	N = 63	IIMs patients showed reduced serum concentrations of 25(OH)-vitamin D compared with HCs. Vitamin D status significantly correlated with muscle enzymes, presence of anti-Jo-1 and anti-Mi-2 antibody, and the absolute numbers of peripheral Treg cells.
Di Luigi, 2013 [19]	N = 20, IIMs, including DM, PM, inclusion body myositis	N = 20	VDR agonists decreased the IFNγ/TNFα-induced CXCL10 protein secretion and targeted cell signaling downstream of TNFα in human skeletal muscle cells of IIMs patients.
Vernerova, 2022 [20]	N = 46 IIMs, including DM, PM and necrotizing myopathy	N = 67	Decreased serum levels of 1,25(OH)D, were observed in IIM patients compared to HCs. Gene expression of VDR and CYP27B1 was numerically higher in the muscle tissue of IIM patients compared to HCs. After a 24-week training program, VDR and CYP27B1 gene expression in primary muscle cells decreased in IIM patients. VDR gene expression in muscle tissue was associated with different disease parameters in IIMs, including myoglobin levels, muscle strength (MMT8) and creatine kinase (CK).
Laiferova, 2021 [21]	N = 7, IIMs	N = 7	IIMs patients showed lower active serum vitamin D compared to HCs before and after training. Active vitamin D levels were positively associated to better fat oxidation in muscle cells. Higher active vitamin D was associated with increased production of energy from fats and better overall fat disposal.
Kapoor, 2011 [33]	N = 362 IIMs including DM, PM and necrotizing myopathy	N = 287	Possession of the A allele in a VDR haplotype tagging SNP (rs2254210) was associated with possession of anti-155/140 antibodies (51%), compared to both controls (33%) (OR 2.1, 95% CI [1.2–3.8], *p* = 0.006) and anti-155/140 antibody negative cases (36%) (1.9, 95% CI [1.1–3.4], *p* = 0.01). Associations for this SNP were observed independently in the juvenile and adult 155/140 (+) sub-groups. No other significant associations were observed in any of the remaining serologically defined sub-groups.
**Non-significant associations**			
Bodoki, 2015 [32]	N = 89 IIMs, including DM and PM	N = 93	There were no differences in either VDR polymorphisms or haplotypes between the IIM patients and healthy subjects

Legend: 25(OH)D: 25-hydroxy-vitamin D; CK: creatin phosphokinase; CXCL10: C-X-C motif chemokine ligand 10; DM: dermatomyositis; HCs: healthy controls; IBM: inclusion body myositis; IFNγ: interferon gamma; IIMs: idiopathic inflammatory myopathies; jDM: juvenile dermatomyositis; MMT8: muscle memory test 8; NA: not assessed; OR: odds ratio; PGA: physician global assessment; PM: polymyositis; TNFα: tumor necrosis factor alpha; VDR: vitamin D receptor; SNP: single nucleotide polymorphism.

## 4. Vitamin D and Polymyalgia Rheumatica

Polymyalgia rheumatica (PMR) is an inflammatory disorder of people over 50 years of age that causes muscle pain and stiffness, especially in the shoulders, neck, and hips [34,35]. It is usually accompanied by an acute phase inflammatory response and its mainstay of treatment are glucocorticoids (GCs) [36,37].

Although muscle histopathologic studies in PMR reveal only minor immunologic abnormalities [38,39], muscle inflammation can occur in the muscle-tendon transition and a recent observational study employing magnetic resonance imaging detected the presence of inflammatory myofascial lesions in all the enrolled new-onset PMR patients (n = 18) [40].

Despite the different PMR incidence across the European Countries, higher in people of Northern European ethnicity [41], there is a relative lack of data about the role of vitamin D insufficiency in predisposing to the pathogenesis of the disease.

Although there are published reports suggesting a seasonal onset of PMR during winter and autumn periods when the cutaneous photosynthesis of vitamin D might be potentially impaired [42,43], a recent meta-analysis carried out by our group failed to document the existence of a significant seasonal pattern for the disease [44]. Similarly a recent study found that active sun exposure, which may increase vitamin D levels, did not affect the risk of developing PMR in women living in southern Sweden [45].

To our knowledge, only a brief report has assessed the serum concentrations of 25 (OH) vitamin D in PMR patients showing an insufficiency (mean serum levels of 24.4 ± 5.7 ng/mL) [22]. However, this study had as a limitation the absence of an age- and sex-matched group of healthy controls. Indeed, several other factors could have contributed to this insufficiency. Elderly age is a significant factor, as the skin’s ability to synthesize vitamin D from sunlight decreases with age [46]. Additionally, low solar light exposure, especially in regions with long winters or in individuals with limited outdoor activity, can further reduce vitamin D synthesis [47]. Malnutrition is another critical factor, particularly in the ‘frail’ population, where inadequate dietary intake can lead to lower vitamin D levels [48]. These factors should be considered when interpreting vitamin D insufficiency in PMR patients, as they may play a crucial role alongside the disease itself.

Moreover, evidence on the genetic aspects, such as polymorphisms in the vitamin D receptor (VDR) and other related genes, is lacking. Understanding whether these genetic factors might influence susceptibility to PMR or vitamin D metabolism in these patients remains an important area for future research. Among the interventional studies assessing outcomes of PMR patients after treatment with vitamin D, a double-blind clinical trial included PMR patients, either randomized to treatment with methylprednisolone (MP) and 25(OH)-vitamin D_3_ (calcifediol) or MP combined with placebo. The assessed outcomes were the disease activity subjectively reported by the patients combined with the serum levels of the erythrocyte sedimentation rate (ESR), mineral metabolism parameters and bone mineral content (BMC). Both groups improved with treatment in terms of disease activity but serum alkaline phosphatase and 24-h hydroxyproline excretion decreased significantly only in the group receiving calcifediol who showed also an increase of the radial BMC compared to the group receiving placebo where it reduced, on the contrary [23].

From these data, the evidence supporting a role in the pathogenesis and a potential modulation of disease activity of vitamin D in PMR patients is insufficient. Therefore, further efforts are warranted in this respect considering the importance of this nutrient in this subset of patients often necessitating treatment with GCs for long-term [49].

## 5. Vitamin D and Sarcopenia in Rheumatoid Arthritis

Rheumatoid arthritis (RA) is a chronic autoimmune disease that primarily affects the synovial joints, causing significant morbidity and impairing the quality of life [50]. While the exact causes are not fully understood, genetic, epigenetic and environmental factors are known to contribute to its pathogenesis. In this respect, vitamin D has emerged as a potential modulator of immune function in RA especially for its ability to induce the activation of tolerogenic dendritic cells and the differentiation of regulatory T and B cells in the synovial microenvironment [51].

Indeed, the majority of studies indicate a potential association between vitamin D deficiency/insufficiency and RA both in terms of risk development for the disease for people showing low serum concentrations [52,53] but also for the inverse correlation between vitamin D status and disease activity in RA patients [54,55,56].

Furthermore, a recent meta-analysis has highlighted the significance of VDR polymorphisms in the susceptibility to RA. This study found that certain VDR gene polymorphisms, such as FokI and TaqI, showed protective associations with RA risk in specific populations such as Europeans, Asians and Arabs while BsmI polymorphism increased RA risk in Africans [57]. These findings suggest that genetic factors related to vitamin D metabolism might also play a role in RA pathogenesis.

Less explored is, instead, the association between vitamin D insufficiency and impaired muscle health in RA which might lead to sarcopenia (loss of muscle mass and strength) in advanced stages of long-standing disease.

A recent observational study has shown, in this respect, that low serum 25(OH)D status was associated with a higher prevalence of severe sarcopenia in older female RA patients [24]. The authors concluded that modification of vitamin D status, including supplementation, should be investigated as a therapeutic strategy for sarcopenic RA patients [24].

Additionally, preliminary results presented at the EULAR Conference in 2022 showed that both vitamin D insufficiency and deficiency were common in RA patients and were associated with poorer locomotive functions, indicating a potential impact on muscle health [25].

In summary, the available evidence indicates that vitamin D insufficiency may be an important factor contributing to impaired muscle health, including sarcopenia, in patients with RA. Further research is still needed to fully elucidate this relationship and determine if vitamin D supplementation could be a beneficial therapeutic strategy.

## 6. Vitamin D, Muscle Strength and Fatigue in Systemic Lupus Erythematosus

In systemic lupus erythematosus (SLE), a chronic autoimmune disease affecting multiple organ systems, vitamin D insufficiency is highly prevalent with studies reporting up to 92% of SLE patients having pathological reduced serum levels [58].

The association between vitamin D status and SLE is complex, with vitamin D deficiency potentially contributing to disease pathogenesis while also being a consequence of the disease process or the prescription of avoid sun light [59]. As a matter of fact, low serum vitamin D levels have been linked to increased disease activity, higher inflammatory markers, and more severe organ damage in SLE patients [60].

Indeed, among contributing factors to vitamin D insufficiency in SLE include the almost total sun avoidance, the use of sun screen and the treatment with GCs which might enhance, especially at medium (>10 mg of prednisone equivalent daily) or high (>25 mg of prednisone equivalent daily) dosages, the catabolism of vitamin D through the activation of the 24-hydroxylase and reduction of calcium absorption [26,61,62].

A previously published meta-analysis has highlighted that certain VDR polymorphisms, such as BsmI and FokI, are associated with an increased risk of SLE, particularly in Asian populations [63]. This suggests that genetic factors related to vitamin D metabolism might also play a role in SLE pathogenesis [63].

Vitamin D supplementation in SLE has shown potential benefits in improving disease activity, fatigue, and autoantibody levels in some studies, but more research is needed to establish optimal dosing and identify patient subgroups most likely to benefit [64].

A study specifically examining the relationship between vitamin D status with fatigue and muscle strength, assessed by hand held dynamometry and tests evaluating the strength of the upper and lower arms, in women with SLE found that fatigue was related to physical function but not to vitamin D status in individuals with SLE [27]. In another recently published study the frequency of sarcopenia was reported in 13% of Turkish SLE patients but vitamin D serum concentrations were not reported highlighting a research gap about this association in SLE [65].

In another mixed cohort of connective tissue diseases (CTDs), including SLE and systemic sclerosis (SSc), patients with vitamin D deficiency demonstrated lower peripheral muscle strength (assessed through supervised knee extension and shoulder flexion exercises), reduced exercise capacity, decreased physical activity, and poorer quality of life compared to those with adequate vitamin D levels. Additionally, these patients experienced significantly higher levels of pain, fatigue, anxiety, and depression (all *p*-values < 0.05) [66].

Interestingly, in a mice model of pristane-induced SLE, treatment with vitamin D positively impacted the muscle physical function by modulating the autophagy mechanisms [67].

A recently published meta-analysis quantitatively summarizing the effect of vitamin D on fatigue in SLE patients provided conflicting results with positive significant effects in terms of standard mean difference and corresponding 95% confidence interval for fatigue when performing exercise, fatigue interfering in social life and the fatigue severity score (FSS) but not significant effects for other domains of this symptom [26].

Therefore, vitamin D supplementation may help in alleviating fatigue in some SLE patients, but more research is needed to establish optimal dosing and identify patient subgroups most likely to benefit.

## 7. Vitamin D and Muscle Health in Other Rheumatic Inflammatory and Autoimmune Connective Tissue Diseases

Among the other autoimmune connective tissue diseases (CTDs), SSc is the disorder displaying the greatest amount of data related to the implication of vitamin D in the pathophysiology of the disease [68].

Interestingly, pre-clinical literature suggests that vitamin D and its analogs may suppress fibrogenesis [69]. The clinical evidence is concordant in reporting a high prevalence of hypovitaminosis D and osteoporosis in SSc patients but the association between clinical manifestations and phenotypes of SSc is less consistent [68].

To our knowledge, only one observational study in SSc has reported a significant inverse correlation between vitamin D serum concentrations and muscle strength assessed clinically [28].

Instead, in other autoimmune CTDs such as Sjögren’s syndrome, mixed connective tissue disease and undifferentiated connective tissue disease, data related to specific muscle health outcomes are scarce and should be a matter of investigation [70,71].

Additionally, recent studies have explored the role of VDR polymorphisms in these diseases. These have been studied in SSc, with some evidence suggesting an association with reduced bone mineral density (BMD) in SSc patients with the homozygous presence of FokI polymorphism [72].

In Sjögren’s syndrome, while VDR gene polymorphisms such as BsmI, ApaI, TaqI, and FokI have been investigated, studies have not found significant associations within the studied populations, suggesting that genetic variations in the VDR gene might not be a major risk factor for developing primary Sjögren’s syndrome in those cohorts [73].

However, the specific impact of these polymorphisms on muscle health outcomes in CTDs remains under-researched.

## 8. Conclusions

This review highlights the multifaceted role of vitamin D in the context of inflammatory and autoimmune rheumatic diseases, with a specific focus on muscle status and health. The literature overview shows that vitamin D insufficiency is prevalent in patients with IIMs, PMR, RA, SLE, SSc and other connective tissue diseases, often correlating with worsened disease outcomes, particularly muscle weakness and sarcopenia.

The evidence suggests that adequate vitamin D serum concentrations may confer protective effects on muscle health by reducing inflammation, enhancing muscle regeneration, improving intra-cellular metabolism and mitochondrial function, also by increasing calcium availability. However, despite these promising associations, there remains a need for more robust longitudinal and interventional studies to confirm these findings and to establish the optimal vitamin D supplementation strategies tailored to specific patient populations.

One area that requires further investigation is determining the optimal serum concentrations of vitamin D necessary to achieve the best muscle-related outcomes in these diseases. In fact, there is also a lack of data on the general outcomes, not just muscle-related, for patients receiving excessive dosages of vitamin D. This gap in knowledge may be due to the relatively rare occurrence of hypervitaminosis D, which, according to a recent study, typically arises from consuming cholecalciferol doses greater than 10,000–40,000 IU per day over several weeks or months [74].

While current evidence supports the potential benefits of maintaining sufficient vitamin D (1,25(OH)) serum concentrations in managing rheumatic diseases and promoting muscle health, further research is essential to delineate the precise mechanisms and therapeutic implications.

## Figures and Tables

**Figure 1 nutrients-16-02329-f001:**
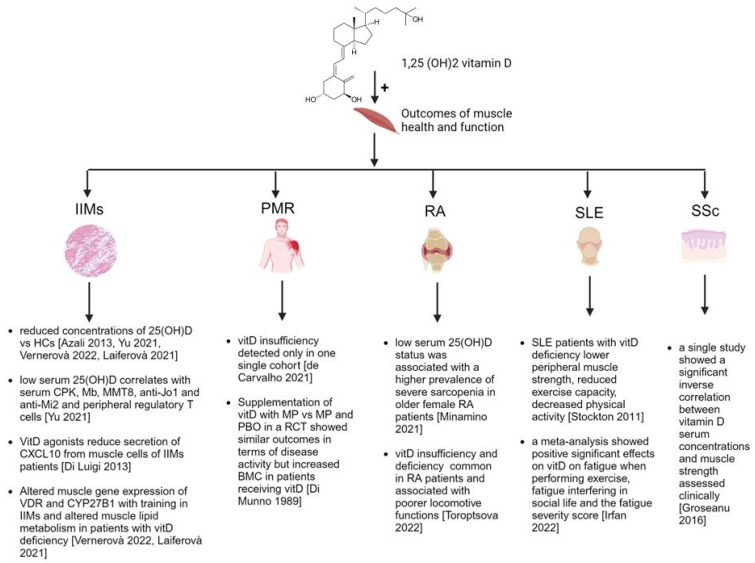
Summarized evidence inherent to vitamin D status and outcomes of muscle health and function in inflammatory and autoimmune rheumatic diseases. Legend: IIMs: idiopathic inflammatory myopathies; PMR: polymyalgia rheumatica; RA: rheumatoid arthritis; SLE: systemic lupus erythematosus; SSc: systemic sclerosis; 25(OH)D: 25-hydroxy-vitamin D; CXCL10: C-X-C motif chemokine ligand 10; CPK: creatin phosphokinase; HCs: healthy controls; MMT8: muscle memory test 8; VDR: vitamin D receptor. References in square brackets inserted in the figure in chronological order of appearance: Azali 2013 [17], Yu 2021 [18], Di Luigi 2013 [19], Vernerovà 2022 [20], Laiferovà 2021 [21], de Carvalho 2021 [22], Di Munno 1989 [23], Minamino 2021 [24], Toroptsova 2022 [25], Irfan 2022 [26], Stockton 2011 [27], Groseanu 2016 [28]. Agreement number of the publication license provided by www.biorender.com: TN272PB7NT.

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
