# Peer review of "Vitamin D and Muscle Status in Inflammatory and Autoimmune Rheumatic Diseases: An Update"

_nutrients, 2024, doi:10.3390/nu16142329_

Round 1

Reviewer 1 Report

Comments and Suggestions for Authors

In this review, Hysa et al approaches vitamin D and muscle status in inflammatory and  autoimmune rheumatic diseases. Although an interesting and

I have a certain criticism concerning the methods of the review [“The results of the search are reported as a narrative review with a focus on the most recent evidence published in the last 5 years.”] I understand that authors focused on recent articles, but I wonder if important studies were not neglected since “last 5 years” seems to be a quite short period.

I am aware that vitamin D plasma levels are important in a wide range of diseases/conditions, but I am also aware of several genetic studies (including cohort of patients attained by different autoimmune rheumatic diseases) suggesting associations between polymorphic variants of genes associated to vitamin D (for example VDR) and the disease development or susceptibility. Actually, authors highlight some genetic studies approaching the VDR gene (for example see manuscript page 4 “…had shown that a specific VDR polymorphism (rs2254210) was more frequently detected in DM …”) but they do not mention if the specific variant was functional or what. I would expect more discussion on those immunogenetic aspects.

In fact, certain VDR polymorphic variants could interfere with an adequately capacity to respond to vitamin D, and lead to “vitamin D deficiency/insufficiency,” even in the presence of “good” vitamin D plasma levels. Could the authors include this point at the discussion?

Also, barely mentioned, are “… other factors (which) might have explained this insufficiency such as elderly age, low solar light exposition, and malnutrition”. I believe such “other factor” can be better discussed at this review.

Finally, nothing is mentioned concerning vitamin D levels and pregnancy. Although I acknowledge that this is not the focus of the present review, the importance of vitamin D on pregnancy should not be neglected.

In conclusion, the manuscript is interesting, and can be enriched specially through the inclusion with a better discussion approaching immunogenetic aspects.

Author Response

Comment 1: In this review, Hysa et al approaches vitamin D and muscle status in inflammatory and autoimmune rheumatic diseases. Although an interesting and

I have a certain criticism concerning the methods of the review [“The results of the search are reported as a narrative review with a focus on the most recent evidence published in the last 5 years.”] I understand that authors focused on recent articles, but I wonder if important studies were not neglected since “last 5 years” seems to be a quite short period.

Response 1: Thank you for your comment. Considering that this review had also the function to provide the readers with an update, the highlight is on the last 5 years, but of course as important background we included and discussed also original papers published in previous years. See Table 1 for instance or the years of the references included in Figure 1. In any case, this has now been rephrased, as you suggest, also in the abstract section of the methods and in the main text (lines 23,24 and 111, 112).

Comment 2: I am aware that vitamin D plasma levels are important in a wide range of diseases/conditions, but I am also aware of several genetic studies (including cohort of patients attained by different autoimmune rheumatic diseases) suggesting associations between polymorphic variants of genes associated to vitamin D (for example VDR) and the disease development or susceptibility. Actually, authors highlight some genetic studies approaching the VDR gene (for example see manuscript page 4 “…had shown that a specific VDR polymorphism (rs2254210) was more frequently detected in DM …”) but they do not mention if the specific variant was functional or what. I would expect more discussion on those immunogenetic aspects.

In fact, certain VDR polymorphic variants could interfere with an adequately capacity to respond to vitamin D, and lead to “vitamin D deficiency/insufficiency,” even in the presence of “good” vitamin D plasma levels. Could the authors include this point at the discussion?

Response 2: Thank you for raising this important point. We have now expanded in the discussion the paragraphs on these immunogenetic aspects of VDR polymorphisms in IIMs in lines 187-192. We added further paragraphs on VDR polymorphisms also including for other diseases if the literature evidence was available but specifying if the associations were related to muscle health outcomes or just related to a higher risk factor for developing the disease in general. Lines 239-242 for PMR, lines 271-276 for RA, lines 307-310 for SLE, lines 355-364 for SSc and Sjogren.

Comment 3: Also, barely mentioned, are “… other factors (which) might have explained this insufficiency such as elderly age, low solar light exposition, and malnutrition”. I believe such “other factor” can be better discussed at this review.

Response 3: Thank you for this comment. In lines 231-238 we have now expanded these concepts adding also new literature references.

Comment 4: Finally, nothing is mentioned concerning vitamin D levels and pregnancy. Although I acknowledge that this is not the focus of the present review, the importance of vitamin D on pregnancy should not be neglected.

Response 4: We agree with you that this is a very important topic but since our review focuses specifically on vitamin D status and muscle health in inflammatory and rheumatic diseases, we believe that it might be potentially out of context in this specific review for the reader and we might not be able to dedicate it the space it would deserve. We hope that you agree with our observation.

Comment 5: In conclusion, the manuscript is interesting, and can be enriched specially through the inclusion with a better discussion approaching immunogenetic aspects.

Response 5: Thank you, we hope that now the revised version of the manuscript as suggested sufficiently covers these aspects.

Most of the time VDR polymorphisms were just mentioned for each disease linked with other health outcomes or as risk factors for developing the disease itself whereas for muscle health outcomes the literature is scanty.

Reviewer 2 Report

Comments and Suggestions for Authors

In this review article, the authors discuss recent progress of vitamin D on muscle health in some of autoimmune diseases. This is an important and interesting topic, which should be of interest to the field. My only suggestion is to improve the language and especially correct grammar errors, only some of which are listed below.

vitamin D is (also) involved in the regulation also of muscle contraction

Although (vitamin D was) traditionally acclaimed for its central role in regulating plasma concentrations of calcium and phosphate preserving a healthy mineralized skeleton, vitamin D it displays many extra-skeletal functions.

such as through reduced use (of vitamin D) due to overall disease activity and secondary sarcopenia.

In Fig.1 a meta-analysis showe (showed)

(also) providing also further immune-pathophysiological insights about vitamin D deficiency in IIMs patients

previous study published as an abstract conference in the UK (an abstract from a conference held in UK) had shown that

might better achieve the control of disease activity (might achieve a better control of the disease)

low solar light exposition (exposure)

As (a) matter of fact

should be (a) matter of investigation

Comments on the Quality of English Language

The language use of the paper needs to be improved.

Author Response

Comment 1: In this review article, the authors discuss recent progress of vitamin D on muscle health in some of autoimmune diseases. This is an important and interesting topic, which should be of interest to the field. My only suggestion is to improve the language and especially correct grammar errors, only some of which are listed below.

vitamin D is (also) involved in the regulation also of muscle contraction

Although (vitamin D was) traditionally acclaimed for its central role in regulating plasma concentrations of calcium and phosphate preserving a healthy mineralized skeleton, vitamin D it displays many extra-skeletal functions.

such as through reduced use (of vitamin D) due to overall disease activity and secondary sarcopenia.

In Fig.1 a meta-analysis showe (showed)

(also) providing also further immune-pathophysiological insights about vitamin D deficiency in IIMs patients

previous study published as an abstract conference in the UK (an abstract from a conference held in UK) had shown that

might better achieve the control of disease activity (might achieve a better control of the disease)

low solar light exposition (exposure)

As (a) matter of fact

should be (a) matter of investigation

Response 1: Thank you for the comments, we revised the manuscript according to your suggestions and English was revised as well. A revised version of Fig 1 was also uploaded to make both the suggested corrections and perform the changes of the references due to the new order.

Reviewer 3 Report

Comments and Suggestions for Authors

The article entitled Vitamin D and Muscle Status in Inflammatory and Autoimmune Rheumatic Diseases: An Update has taken into consideration the role of Vit D in different kinds of autoimmune diseases. What is positive they discuss the role of vitamin D in

 idiopathic inflammatory myopathies, polymyalgia rheumatica, rheumatoid arthritis, systemic lupus erythematosus, systemic sclerosis. Influence of Vit D deficiency on the above "diseases" induction and therapeutic role by nutrition fortification. From the diet supplementation market point, it is positive, however, the lack of information about the consequences (negative) of Vit D overload is unacceptable.

Due to the above, I cannot recommend this article in its present form.

The Nutrients journal is open and accessible to broad audiences not only scientific.

Moreover, the article looks like slightly connected data without deep scientific bedground (what is well visible in the references), even if it is readable. I recommend to make more effort for the editorial work.

Therefore, I strongly recommend authors introduce the negative points of Vit D supplementation, and during the diseased and physiological defect discussion make the effort and describe the mechanism of Vit D action.

Author Response

Comment 1: The article entitled Vitamin D and Muscle Status in Inflammatory and Autoimmune Rheumatic Diseases: An Update has taken into consideration the role of Vit D in different kinds of autoimmune diseases. What is positive they discuss the role of vitamin D in idiopathic inflammatory myopathies, polymyalgia rheumatica, rheumatoid arthritis, systemic lupus erythematosus, systemic sclerosis. Influence of Vit D deficiency on the above "diseases" induction and therapeutic role by nutrition fortification. From the diet supplementation market point, it is positive, however, the lack of information about the consequences (negative) of Vit D overload is unacceptable.

Due to the above, I cannot recommend this article in its present form.

The Nutrients journal is open and accessible to broad audiences not only scientific.

Moreover, the article looks like slightly connected data without deep scientific bedground (what is well visible in the references), even if it is readable. I recommend to make more effort for the editorial work.

Therefore, I strongly recommend authors introduce the negative points of Vit D supplementation, and during the diseased and physiological defect discussion make the effort and describe the mechanism of Vit D action.

Response 1: Thank you for your detailed feedback on our manuscript.

We appreciate your insights and have made several revisions to address your concerns.

We have expanded the introduction to include detailed information on the immunomodulatory properties of vitamin D, as per your suggestion (new lines 62-71). This addition provides a deeper scientific background on how vitamin D influences both innate and adaptive immune responses.

We acknowledge also the importance of discussing the potential negative effects of excessive vitamin D supplementation. In the conclusion section, we have included information on hypervitaminosis D and highlighted the need for further research in this area (new lines 376-382)

“In fact, there is also a lack of data on the general outcomes, not just muscle-related, for patients receiving excessive dosages of vitamin D. This gap in knowledge may be due to the relatively rare occurrence of hypervitaminosis D, which, according to a recent study, typically arises from consuming cholecalciferol doses greater than 10,000-40,000 IU per day over several weeks or months[73].”

As our review specifically targets muscle health in autoimmune and inflammatory rheumatic diseases, we have noted that data specific to muscle outcomes are more limited compared to general disease activity or autoimmune risk.

We hope these revisions meet your expectations and provide the coverage you suggested.

Round 2

Reviewer 1 Report

Comments and Suggestions for Authors

Authors included several paragraphs approaching the potential involvement of VDR gene variants (as well as f the VDR receptor molecule) in autoimmune diseases. Therefore, the manuscript was enriched. They also included a paragraph addressing other factors that could affect Vitamin D levels. Finally, I agree with their decision do not include a discussion about the effect of Vitamin D in pregnancy, since I understand that this is not the focus of the present review.

In this sense, I believe that the manuscript is in condition to be published.

Reviewer 3 Report

Comments and Suggestions for Authors

The authors have provided the correct answers to my questions. However, the answers are not so extended in detail.

Therefore at present form, the article can be accepted.